# Carbon-Based Materials in Photodynamic and Photothermal Therapies Applied to Tumor Destruction

**DOI:** 10.3390/ijms23010022

**Published:** 2021-12-21

**Authors:** Karina J. Lagos, Hilde H. Buzzá, Vanderlei S. Bagnato, María Paulina Romero

**Affiliations:** 1Department of Materials, Escuela Politécnica Nacional (EPN), Quito 170525, Ecuador; karina.lagos@epn.edu.ec; 2Institute of Physics, Pontificia Universidad Católica de Chile, Santiago 7820436, Chile; hilde.buzza@fis.uc.cl; 3São Carlos Institute of Physics, University of São Paulo (USP), São Carlos 13566-590, Brazil; vander@ifsc.usp.br

**Keywords:** phototherapy, cancer, graphene oxide, reduced graphene oxide, graphene quantum dots, carbon dots

## Abstract

Within phototherapy, a grand challenge in clinical cancer treatments is to develop a simple, cost-effective, and biocompatible approach to treat this disease using ultra-low doses of light. Carbon-based materials (CBM), such as graphene oxide (GO), reduced GO (r-GO), graphene quantum dots (GQDs), and carbon dots (C-DOTs), are rapidly emerging as a new class of therapeutic materials against cancer. This review summarizes the progress made in recent years regarding the applications of CBM in photodynamic (PDT) and photothermal (PTT) therapies for tumor destruction. The current understanding of the performance of modified CBM, hybrids and composites, is also addressed. This approach seeks to achieve an enhanced antitumor action by improving and modulating the properties of CBM to treat various types of cancer. Metal oxides, organic molecules, biopolymers, therapeutic drugs, among others, have been combined with CBM to treat cancer by PDT, PTT, or synergistic therapies.

## 1. Introduction

Phototherapy is a non-traditional strategy that has been used within several bio-applications. For example, in antimicrobial treatments, light stimulation of an agent promotes the inactivation of bacteria, protozoa, viruses, and fungi [1,2,3]. Likewise, various diseases such as vitiligo [4], psoriasis [5], atopic dermatitis [6], cancer [7], and so on, have been diagnosed and treated by this approach.

Cancer has become a disease of significant concern in recent years due to its threat to human life, causing millions of deaths. The International Agency for Research on Cancer reported 9.9 million worldwide diseases in 2020 (world ASR of 100.7) [8]. Thus, several studies have proposed phototherapy using nanomaterials as photoabsorbing agents as an alternative to treat cancer [9,10,11]. It is worth noting that phototherapy is advantageous compared to radiotherapy, chemotherapy, or surgery owing to its simple operation, minimally invasive procedure, reduced toxicity, minor trauma, fewer adverse reactions, and negligible drug resistance [12,13]. Nevertheless, it has drawbacks as poor penetration limits its action in optically inaccessible deep tumors. To overcome these weaknesses, novel flexible light sources and devices have been designed, and approaches such as X-ray radiation, NIR light, and internal self-luminescence have been proposed [14].

Photodynamic therapy (PDT) and photothermal therapy (PTT) are encompassed within phototherapy. When light of a suitable wavelength is irradiated upon a defined molecule photosensitizer (PS) or photothermal agent (PA), reactive oxygen species (ROS) and heat are generated, respectively, in PDT and PTT, which causes damage to malignant cells in cancer [15,16]. In PDT, the PS is excited by light. In this state, the PS reacts with nearby molecular oxygen and generates ROS, either type I (free radicals) or type II (singlet oxygen, ^1^O_2_) reactions. In cancer treatments, this action results in apoptosis, necrosis, or autophagy of the abnormal cells inhibiting the tumor growth [17]. The efficiency of PDT is related to the ROS generation yield, which depends on the PS, dose, source light, and tissue oxygen [18]. PTT is based on localized hyperthermia. The PAs are irradiated by light and they absorb photons which produces an excited state. By non-radiative relaxation pathways, heat is generated to dissipate this excess of energy [19]. When the temperature of the PAs surrounding the environment rises, cancerous cells are destroyed due to their low heat tolerance compared to normal cells. Thus, the targeting capability of PSs and PAs in tumor cells is key to concentrating their action in cancer tissue [20,21].

In both therapies, PDT and PTT, particular properties in their active agents are required, as well as robust responses to light stimuli. Furthermore, high specificity, biocompatibility, low dark toxicity, and optical characteristics are desirable [22]. On this basis, carbon-based materials (CBM) have become excellent candidates as phototherapy agents and as platforms or carriers of these compounds [17,23,24,25]. The specificity of CBM accomplishes focusing its action only on cancerous cells. These materials are critical components due to their remarkable advantages, such as reduced side effects and low toxicity in specific concentrations [17,26]. Phototherapy agents can also be loaded with drugs or combined with other materials to enhance their antitumoral action or to improve and modulate their properties, making possible the effective application of distinct mechanisms of action [27]. Thus, the doping and hybrid behavior of CBM along with synergistic therapies are also addressed in this review.

## 2. Carbon-Based Materials (CBM) Applied in Photodynamic (PDT) and Photothermal (PTT) Therapies

CBM have been considered a phototherapy agent due to their remarkable features. Nevertheless, it is worth noting that the properties of any CBM vary according to their specific structure (size and shape), which is determined by the method of synthesis, along with their experimental conditions and carbon source nature. Concerning the latter, some authors have even proposed using residues such as bio-mass and polymers waste [28,29]. To synthesize CBM, two approaches have been employed; (i) top-down: reduction of size from bulk materials, such as mechanical or chemical exfoliation, and (ii) bottom-up: construction from the atomic level, like epitaxial growth and chemical vapor deposition [30,31].

Within CBM, graphene has become one of the most studied materials owing to its unique chemical and physical properties [32,33] which have encouraged its application in diverse fields such as electronics, material science, energy, and biomedicine, including the treatment of the COVID-19 disease [34,35,36,37]. Regarding bio-applications, the versatility of graphene has allowed its assessment as an antimicrobial agent [38], in sensors [39], drug delivery [40], bioimaging [41], regenerative medicine [42], cancer treatment [43,44], and photodynamic and photothermal therapies [45,46]. However, the hydrophobicity of the pristine graphene has turned into a drawback when affinity with physiological solutions or water is desirable [47]. Moreover, graphene tends to agglomerate in solution and has poor solubility. In this context, some alternatives have been proposed to overcome this limitation. For example, functionalized graphene derivatives such as graphene oxide (GO), reduced graphene oxide (r-GO), or graphene quantum dots (GQDs) have been employed instead (see Figure 1). These derivatives have excelled as novel materials due to their large specific surface area, bio-compatibility, solubility, and selectivity [48,49].

Another enthralling new material with comparable properties to those of graphene derivatives is carbon dots (C-DOTs). These 0D materials with sizes below 10 nm and an easy ability to be synthesized have also been extensively used in bio-applications [51,52,53].

In this section, GO, r-GO, GQDs, C-DOTs, and their composites and hybrids are addressed as new materials in phototherapy and synergistic therapies against cancer disease.

### 2.1. Graphene Oxide (GO)

GO has a 2-dimensional (2D) honeycomb structure with sp^3^ domains enclosing sp^2^ carbon domains and is covalently functionalized with carbon and oxygen groups [54]. These attachments differentiate GO from graphene. The presence of carboxylic (-COOH), hydroxyl (-OH), carbonyl (C=O), alkoxy (C-O-C), epoxy (>O), or other functional groups induces changes in specific GO properties such as its characteristic insulator behavior and water affinity [55,56]. The functionalizations allow the establishment of covalent bonds with other species. Figure 2 presents some materials that have been employed to modify GO’s surface.

Some studies have reported sulfur contents in GO, which also induces substantial variations in its acidic and electric properties [57]. It is worth indicating that these couplings trigger an expansion in the interlayer spacing, doubling the GO size compared to graphene. Generally speaking, the properties of GO consist of the planar layer structure, high thermal and electrical conductivity, excellent optical transmittance, and flexibility of surface modification [58].

**Figure 2 ijms-23-00022-f002:**
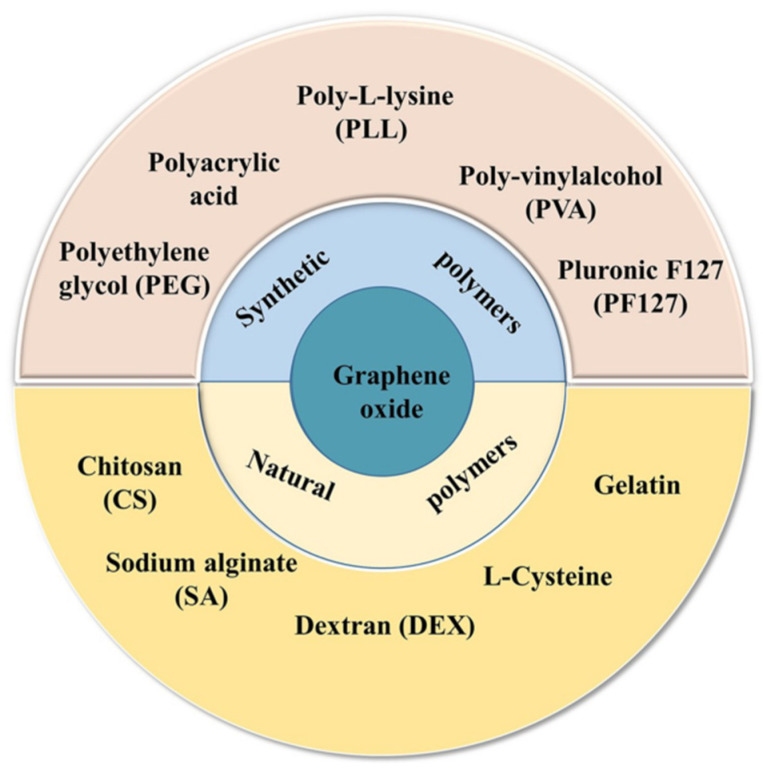
Materials used for the surface modification of GO. Reproduced from Ref. [59]. Copyright 2021 Springer Nature.

#### 2.1.1. Application of GO in PDT

Several studies have employed GO and standard photosensitizers (PSs) for cancer treatment. Hosseinzadeh R. et al. (2018) fabricated a PS using GO and methylene blue. The potential of the PS for killing cancer cells was evaluated by the Thiazolyl Blue (MTT) cell viability assay employing a human breast cancer cell line (MDA-MB-231). Using a concentration of 20 µg/mL in dark conditions, the results of cell viability showed a reduction of up to 60% for the PS, having a much better performance than the components separately, which did not present a significant decrease, barely less than 5%. In contrast, under irradiation with red LED illumination of 630 nm for 30 min, a reduction of up to 80% was obtained employing the same concentration of the PS [60]. Likewise, Sun X. et al. (2018) developed a GO-based nanocomposite. For this, they encapsulated TPE-red (tetraphenylethylene- aggregation-induced emission nanoparticles) with modified GO by PEGylation procedure. They demonstrated that the nanocomposite increased the production of ROS under laser irradiation of 450 nm, enhancing the ROS generation capability compared to single TPE-red. Besides, an MTT assay was carried out on UMUC3 cells indicating higher toxicity under radiation than in dark conditions [61]. Qin et al. (2018) fabricated a nanocomposite with GO, magnetic nanoparticles (Fe_3_O_4_), chitosan, and a novel photosensitizer HNPa (3-[1-hydroxyethyl]-3-divinyl-131-b,b-dicyano-methylene-131-deoxopyropheophorbide-a). They confirmed a superior singlet oxygen quantum yield compared to the single HNPa, being 62.9% and 42.6%, respectively. Furthermore, they demonstrated that the presence of GO-Fe_3_O_4_ accelerated the penetration of HNPa into the nucleus of the human hepatocellular carcinoma cell line (HepG-2). Besides this, an MTT assay carried out under 698 nm of irradiation verified an enhanced result of HNPa to increase photodynamic cancer cell death [62].

A simplified scheme of GO-composite application in PDT is shown in Figure 3.

#### 2.1.2. Application of GO in PTT

The excellent efficiency of GO photothermal conversion in the NIR region makes it susceptible to use in PTT [64]. In recent years, highly elaborate hybrids have been developed to improve the solubility and selectivity of GO. Lim et al. (2018) fabricated a ~155 nm nanocomposite using GO along folic acid and manganese dioxide (MnO_2_). In cancer, the MnO_2_ decomposes hydrogen peroxide into oxygen, relieving hypoxia. The results show that the composite heat capacity was better than a single GO. Under 808 nm laser excitation of 3.5 min, the nanocomposite reaches the desired temperature of 47 °C while GO reaches barely 35 °C [65].

Similarly, Xie et al. (2019) fabricated a composite with good stability and dispersibility from GO, magnetic nanoparticles (Fe_3_O_4_), chitosan, sodium alginate, and doxorubicin hydrochloride (DOX). They verified the composite PTT properties using an MTT assay with human lung cancer cell line (A549) and irradiation of 808 nm for 5 min, demonstrating excellent intracellular uptake characteristics and dependence of the increase in temperature with the concentration. The best result shows a reduction of the survival rate to 14.36% with a dose of 100 μg/mL [66]. Furthermore, in the study by Huang and coworkers (2019), they developed a composite with indocyanine green (IR820), lactobionic acid (LA), DOX, and GO. They compared the photothermal capabilities of the composite and single GO, verifying a better performance of the composite due to an increase in temperature of 16.6 °C and 8.2 °C, respectively, after 5 min of irradiation of 660 nm [67]. These results can be seen in Figure 4.

#### 2.1.3. Application of GO in Synergistic Therapy

A synergistic effect is the result of two or more processes interacting to produce a greater action than the individual ones. The assembly of composites through different materials with complementary properties allows its application in synergistic therapy. Several authors have studied the PDT/PTT synergistic effect of GO-based nanocomposites [68,69]. Gulzar et al. (2018) fabricated a hybrid with GO, amino-modified upconversion nanoparticles (NaGdF_4_:Yb^3+^/Er^3+^@NaGdF_4_:Nd^3+^/Yb^3+^), polyethylene glycol (PEG), and Chlorin e6 (Ce6). Singlet oxygen generation was confirmed through the DPBF (1,3-diphenyliso-benzofuran) chemical probe (PDT effect). In addition, an in vivo antitumor property was evaluated in mice using a U14 (murine hepatocarcinoma) cell line with irradiation of 808 nm. As a result, the variation in the relative volume of the tumor (V/V_o_) was reduced by half after 14 days as a consequence of the PTT effect, while this value in the control group increased by nine times [70]. Zhang et al. (2019) tested PDT, PTT, and chemotherapeutic effects of a composite fabricated with GO, wedelolactone, and indocyanine green. Under NIR irradiation (808 nm), ROS (singlet oxygen) generation was confirmed by DCFH-DA probe in human cervical carcinoma cells (HeLa cells). Moreover, stable and the reaching of high heat levels were proved (~79 °C) in comparison to the individual components (~33 °C) [71]. Romero et al. (2021) functionalized the GO surface modified with PEG-folic acid, rhodamine B, and indocyanine green to treat Ehrlich tumors in mice by in vivo experiments using PDT and PTT with an NIR light of 808 nm 1.8 W/cm^2^. Based on fluorescence images of the tumor, the highest concentration of GO as a function of the time after intraperitoneal injection was determined [72].

A basic scheme of the procedure to apply GO in synergistic therapy is shown in Figure 5.

### 2.2. Reduced Graphene Oxide (r-GO)

r-GO is a material that exhibits excellent graphene-like properties. Generally, it is obtained from GO by some methods. Thermal, chemical, photocatalytic, laser, and electrochemical treatments, among others, have been developed to carry out the reduction from GO to r-GO [73]. Figure 6 presents the thermal reduction of GO to r-GO at different temperatures. In these procedures, functional groups are removed from the GO surface, and a structure similar to graphene is achieved with some imperfections and different magnitudes. The C/O ratio is the primary indicator used to verify the quality of the GO reduction [56,74].

Several applications such as catalysts, membranes, super-capacitors, flexible sensors, bio-applications, and more, are based on r-GO [75]. Regarding the latter, specifically in phototherapy, r-GO is considered an excellent PS due to its ability to absorb visible and near-infrared ranges over the whole spectrum. Within cancer disease treatment, r-GO can be functionalized using conjugated molecules that can be employed as a drug delivery platform. Thus, r-GO has been employed in some therapies against cancer, as shown in Figure 7.

#### 2.2.1. Application of r-GO in PDT

Vastly elaborate hybrids have been developed in the last few years to improve the r-GO solubility and selectivity in PDT applications. Thus, Kapri and Bhattacharyya (2019) synthesized a composite with nitrogen-doped r-GO, molybdenum sulfide, manganese dioxide, and PEG to test its photodynamic properties against cancer. They employed the MTT assay with HeLa cells and human embryonic kidney (HEK 293) cells. With a high dose of 200 µg/mL of the composite, the cell viability reached ~85% for both cases. They used NIR irradiation of 980 nm, achieving a disproportionation production of intracellular H_2_O_2_ turning this composite to an enhanced PS [77]. Likewise, Vinothini and coworkers (2020) developed an r-GO-based composite with magnetic nanoparticles (Fe_3_O_4_), camptothecin, 4-hydroxycoumarin, and allylamine, to evaluate its photodynamic capability. They demonstrated high ROS generation and consequently good inhibition against MCF-7 cells (human breast cancer cell lines) under 365 nm of laser irradiation [78]. Green approaches have also been proposed for this. Jafarirad et al. (2018) fabricated three hybrids with r-GO, zinc oxide (ZnO), neodymium (Nd), and silver (Au) nanoparticles (ZnO/r-GO, Nd-ZnO/r-GO, and Ag-ZnO/r-GO) using a rosehip extract (*Rosa canina* L.) as a stabilizing and reducing agent. They satisfactorily demonstrated its antitumor capability employing 630 and 810 nm wavelengths of irradiation [79].

#### 2.2.2. Application of r-GO in PTT

r-GO has a good photothermal conversion response and can effectively produce an overheating effect when used with IR light. Lima-Souza et al. (2018) developed an r-GO nanocomposite using hyaluronic acid and poly maleic anhydride-alt-1-octadecene. The hyaluronic acid was carefully chosen due to its hydrophilic behavior and targeting capacity to CD44 receptors in the cancer cell’s membrane. This formulation presented heat capacity since, after NIR irradiation, its temperature increased to 33 °C, which induced the cancer cell’s death. In addition, enhanced cytocompatibility and stability were achieved in comparison to a single r-GO [80]. Likewise, a different light source as low-intensity LED has been tested in PTT. De Paula et al. (2020) employed red LED (640 nm) ablation to significantly decrease the tumor mass of mice (melanoma in B16F10 lineage cells) using an rGO-based treatment. On the first day, the mean volume of the tumor was ~70 mm^3^, and after 8 days of treatment, it reduced to ~40 mm^3^. Moreover, the immune response was verified by detecting the growth in CD8+ T cells [81]. Liu et al. (2019) developed an elaborate hydrogel with carboxymethyl chitosan (CMC), r-GO, aldehyde (CHO), and PEG (see Figure 8). They assessed its photothermal effect and controllable DOX release under 808 nm of irradiation. The composite’s thermal behavior showed a notable improvement due to the presence of r-GO since the temperature increased from 25 °C to ~65 °C after exposure to light, while, prescinding this CBM, the temperature only reached ~33 °C [82].

#### 2.2.3. Application of r-GO in Synergistic Therapy

Concerning synergist therapy, Zaharie-Butucel et al. (2019) combined PDT, PTT, and chemotherapy using a composite of r-GO, chitosan, IR820, and DOX. Its anticancer activity was satisfactorily assessed by cell proliferation assay against C26 cells (murine colon carcinoma) under NIR irradiation of 785 nm. In addition, the authors demonstrated that the composite penetrated the cytoplasm and the nucleus using scanning confocal Raman microscopy. The composite was the tracker of the living cells, owing to the underlying lattice of the r-GO [83]. Wang et al. (2020) studied photothermal action and used an r-GO/PEG-NH_2_/Fe_3_O_4_ composite under irradiation of 805 nm to eliminate primary tumors. Moreover, this nanomaterial encouraged antitumor immunity [84]. Likewise, Wei G. et al. (2016) evaluated PDT and PTT characteristics of an elaborate composite fabricated from r-GO, graphene diazotized, polyethylenemine, and tetrakis (4-carboxyphenyl) porphyrin against CBRH7919 cancer cells inducing apoptosis after irradiation due to singlet oxygen and heat generation [85]. Zhang et al. (2017) treated A549 lung cancer cells using two different light sources of 808 nm and 450 nm to PTT and PDT, respectively. As PS, they used a composite based on r-GO along with PEG-modified Ru (II) complex. This combination resulted in better cytotoxicity and an enhanced reduction in tumor volume, which was observed from in vivo tests. The PTT–PDT treatment inhibits the growth of the tumor, reducing the relative tumor volume value (V/V_o_) close to zero, in contrast to the PTT and PDT alone, which showed an increase in this value to ~1.5 and ~2.5, respectively [86]. The results of the mentioned work are presented in Figure 9.

### 2.3. Graphene Quantum Dots (GQDs)

The zero-dimensional GQDs are emerging graphene derivatives. Their thickness, less than 100 nm, consists of a maximum of 10 stacked layers of graphene sheets. These reduced dimensions trigger quantum confinement and special edge effects [53,87]. Due to the low toxicity, biocompatibility and photostability, GQDs have many applications such as cell imaging, drug carrier, biosensors, and so on [88].

In order to expand the narrowed visible photoluminescence of GQDs to all visible and infrared, nitrogen doping has been considered [89]. GQDs have also been employed in several phototherapy studies [90,91], including PDT and PTT. GQDs were used as single photo absorbing agents or in the development of composites [92].

A scheme of the inherent effects, preparation methods, properties, and applications of GQDs is presented in Figure 10.

#### 2.3.1. Application of GQDs in PDT

Ge et al. (2014) fabricated GQDs from 2 nm to 6 nm in diameter, which showed ROS generation capability and was employed within in vivo experiments. After 9 and 17 days of treatment of female BALB/c mice with GQDs, a reduction of the tumor was verified [93]. Tabish et al. (2018) synthesized 20 nm GQDs with a 7.1% yield and demonstrated its ROS generation under irradiation of 365 nm. In addition, limited toxicity was verified by in vitro and in vivo tests [94]. Campbell et al. (2021) developed a nanocomposite based on three covalently bounded components: nitrogen-doped GQDs, hyaluronic acid, and ferrocene. The composite did not present a significant cytotoxic response at concentrations of up to 1 mg/mL for HEK-293 cells, greater than 90% in cell viability. In contrast, the composite action against HeLa cells promoted better cytotoxicity, up to 20% after 72 h. Moreover, therapeutic ROS generation was three times higher than that of single ferrocene [95]. It is worth noting that the performance of GQDs in PDT might be associated with its specific structural features. Chen et al. (2020) examined the photoactivity of single-atomic-layered GQDs under laser and halogen irradiation, demonstrating a null generation of ^1^O_2_, which was attributed to its particular morphology [96].

On the other hand, regarding the doping of GQDs, it has been considered to improve its phototherapy performance. Elements like sulfur and nitrogen have been employed to achieve this aim. Concerning the latter, some studies have determined that nitrogen-bonding increased ROS generation compared to single GQDs [97,98].

#### 2.3.2. Application of GQDs in PTT

PTT studies have also evaluated irradiations below the second NIR window (1000–1700 nm) to improve penetration and enhance the damage of the tumor. As a result, Liu et al. (2020) fabricated 3.6 nm GQDs and assessed their photothermal properties under 1064 nm wavelength. In vitro and in vivo tests demonstrated that GQDs killed tumor cells and inhibited tumor growth, respectively [99]. Yao et al. (2017) studied the heat generation of magnetic mesoporous silica nanoparticles capped with GQDS under an alternating magnetic field and NIR irradiation. This material showed efficient PTT and magnetic hyperthermia in in vitro experiments [100]. Wang et al. (2019) synthesized GQDs doped with nitrogen and boron and analyzed the composite under NIR-II region. In vitro and in vivo tests demonstrated the photothermal effect using a glioma xenograft mouse model [101]. Li et al. (2017) loaded IR780 dye on folic acid (FA) functionalized GQDs and studied their behavior under irradiation of 808 nm for 5 min. The temperature of a mice tumor was raised to 58.9 °C, and in vivo antitumor experiment presented a suppressive effect on tumor growth, dissipating it in 15 days, as can be seen in Figure 11 [102].

#### 2.3.3. Application of GQDs in Synergistic Therapy

Synergistic therapy has also been studied employing GQDs. Wang and coworkers (2020) developed a composite with cRGD (Cyclic Arg-Gly-Asp peptide) and DOX to evaluate its photothermal activity against SK–mel–5 and H460 cells under NIR irradiation of 808 nm. Furthermore, its chemotherapy capacity was verified, demonstrating an IC50 reduction up to 39.63 μg/mL and 53.75 μg/mL, respectively [103]. Likewise, Zheng et al. (2019) developed a composite with GQDs, DOX, and hollow copper sulfide nanoparticles within photothermal-chemotherapy applications using NIR irradiation on MDA-MB-231 cells. This research demonstrated a high therapeutic effect in tumor cells and its potential in cancer therapy [104]. To evaluate its NIR response, Thakur et al. (2017) produced GQDs using waste as a carbon source (see Figure 12), specifically withered leaves of Ficus racemose (Indian fig tree). They demonstrated that GQDs were cytocompatible employing cell cycle analysis by flow cytometry and biocompatibility studies. Moreover, it was demonstrated that upon irradiation of 808nm wavelength (0.5 W cm^−2^), the concentration dependence of photothermal response and the production of reactive oxygen species were achieved [105].

### 2.4. Carbon Dots (C-DOTs)

C-DOTs can be generally defined as a quasi-0D carbon-based material with sizes below 10 nm. These materials have spherical or hemispherical structures corresponding to a core of carbon atoms with hybridization inter sp^2^ whose surface is functionalized [106,107]. C-DOTs, new rising stars in the carbon family, have attracted substantial attention due to their excellent and tunable photoluminescence, high quantum yield, fluorescence, low toxicity, small size, appreciable biocompatibility, and abundant, low-cost sources, providing essential applications in many fields, including biomedicine, catalysis, optoelectronic devices, and anticounterfeiting [108,109,110].

C-DOTs have been deemed appropriate for phototherapy applications, not only as photo absorbing agents but also as nano-carriers. Several hybrids obtained by covalent coupling, electrostatic interaction, or π-π stacking have been tested in recent years [111,112].

#### 2.4.1. Application of C-DOTs in PDT

The antitumor effect of C-DOTs conjugates within PDT to treat cancer disease has been confirmed by several authors. Li et al. (2017) prepared porphyrin-containing C-DOTs from mono-hydroxylphenyl triphenylporphyrin and chitosan through a simple one-pot hydrothermal method. They evidenced the effective photodynamic activity toward human hepatocellular liver carcinoma (HepG2) cells by MTT assay under LED irradiation of 625 nm for 1 h. With a concentration of 0.5 mg/mL, cells presented lysis, dramatic apoptosis, and membrane disruption. Their results show that the material possesses good photostability, biocompatibility, cellular uptake, and potent cytotoxicity upon irradiation. In the in vivo test, the size of the irradiated mice tumor was reduced from 100 to 56 mm^3^, while without irradiation, an increment in the tumor size to ~800 mm^3^ was detected [113]. Huang et al. (2012) prepared a novel theranostic system based on Ce6-conjugated C-DOTs. The in vitro results determine that the composite upon irradiation exhibits good stability and solubility, low cytotoxicity, good biocompatibility, enhanced photosensitizer fluorescence detection, and remarkable photodynamic efficacy compared to Ce6 alone. Furthermore, the in vivo results suggest that the newly synthesized nanocomposite possesses excellent imaging efficacy [114]. Qin et al. (2021) produced C-DOTs by microplasma using o-phenylenediamine, revealing a broad absorption peak at 380–500 nm and emitted bright yellow fluorescence with a peak at 550 nm. The C-DOTs were rapidly taken up by HeLa cancer cells. A bright yellow fluorescence signal and intense ROS were efficiently produced when excited under blue light, enabling simultaneous fluorescent cancer cell imaging and photodynamic inactivation, with a 40% decrease in relative cell viability [115].

A simplified scheme of the production of ROS for PDT based on C-DOTs is shown in Figure 13.

#### 2.4.2. Application of C-DOTs in PTT

The study by Meena and coworkers (2019) demonstrates that C-DOTs, synthesized from ayurvedic medicinal plants, with a concentration of 0.5 mg/mL and under 10 min of light exposure (750 nm), reached a temperature of up to 46 °C, verifying its potential on photothermal therapy. Furthermore, no significant toxicity was revealed against NIH-3T3 normal cells, indicating attractive behavior in this field [117]. Sun et al. (2016) demonstrated that red emissive C-DOTs were able to efficiently and quickly convert laser energy into heat and that upon laser irradiation for 10 min, the viability of MCF-7 cells was significantly reduced as the concentration increased (20–200 μg/mL) [118]. Geng et al. (2018) showed that NIR-absorbing nitrogen and oxygen co-doped C-DOTs generated high-efficiency heat under laser irradiation at a low power density achieving 100% of tumor ablation without causing any side effects [119]. Zheng et al. (2016) synthesized an NIR fluorescent composite (from 600 nm to 900 nm) from a hydrophobic cyanine dye and PEG with preferential uptake and accumulation to tumors and high photothermal conversion efficiency (38.7%) as a novel theranostic agent for NIR fluorescent imaging and PTT in vivo and in vitro [120]. Likewise, Li et al. (2019) developed a novel second near-infrared (NIR-II) emitting C-DOTs which had good photothermal efficiency of up to 30.6% under irradiation of 808 nm. The detailed results of this work are presented in Figure 14 [121].

#### 2.4.3. Application of C-DOTs in Synergistic Therapy

Jia et al. (2018) tested in vivo/in vitro PDT and PTT properties of green synthesized C-DOTs against HeLa cells under 635 nm of irradiation, demonstrating 0.38% of quantum yield and 27.6% of photothermal conversion efficiency [122]. It is worth noting that ROS and heat generation are desirable, but selectivity is essential. For example, besides cancer cells, lysosome targeting has also been assessed, showing promising results [123]. Furthermore, some studies have been published comprising synergistic therapy with multifunctional C-DOTs. Consequently, Lan et al. (2018) synthesized C-DOTs and confirmed their PDT and PTT activities under 800 nm of irradiation. In addition, their fluorescence and photoacoustic properties for imaging were verified [124]. Similar evaluations were carried out by Sun et al. (2019) using amino C-DOTs modified with 0.56% (*w*/*w*) of Ce6, but with irradiation of 671 nm [125]. In this context, Guo and coworkers (2018) developed Cu, N-doped C-DOTs using different temperatures in a simple hydrothermal method. They evaluated the PDT and PTT capabilities under 808 nm, determining ROS generation and a rise in temperature of up to 53 °C [126]. Zhang et al. (2018) synthesized a therapeutic agent from DOX loaded in a conjugated sgc8c aptamer (5′-NH2-TGA ATG TTG TTT TTT CTC TTT TCT ATA GTA-3′), with single-walled carbon nanotubes (SWCNTs), PEG, Fe_3_O_4,_ and carbon quantum dots with a multifunction ability that can target and kill cancer cells by releasing the drug photodynamically or photothermally. The composite converted 808 nm NIR into heat energy, generated ROS, and removed cancer cells. These nano-carriers are favorable for treating cervical cancer and other diseases that need precise drug targeting [120]. Yang et al. (2019) synthesized C-DOTs/Hemin. The composite could increase the temperature enhancement to 26 °C under laser irradiation with outstanding photodynamic efficacy. More than 90% of cancer cells die after 10 min laser treatment. This hybrid showed high ROS generation using the DCFH-DA probe against HepG2 cells [127].

A simplified scheme of ROS production for PDT and heat for PTT therapies based on C-DOTs is shown in Figure 15.

## 3. Advantages of CBM in PDT and PTT over Metal-Organic and Organic PSs

The advantages of CBM have been assessed by a number of authors. Ge et al. (2014) evaluated the PDT efficiency and cytotoxicity against HeLa cells under irradiation of 630 nm for 10 min using GQDs and protoporphyrin IX (PpIX). Employing 0.036 µM of GQDs, a cell viability of 60% was detected. This parameter was reduced when the GQD concentration was increased. Nevertheless, the survival of HeLa cells was not greatly influenced by GQDs, indicating its good biocompatibility and low cytotoxicity. In contrast, using 1.8 µM of PpIX, a low cell viability of 55% was achieved in the dark conditions, and more than 35% of the cells survived even upon irradiation. GQDs showed superior PDT efficiency compared to PpIX and lower cytotoxicity [93].

Ge et al. (2016) prepared C-DOTs using polythiophene benzoic acid as a carbon source. The degradation of disodium 9,10-anthracendipropionic acid (Na_2-_ADPA) was confirmed by the generation of singlet oxygen from C-DOTs and the PS methylene blue. The quantum yield of the C-DOTs was calculated to be 0.27, and the MB was 0.58 in CD_3_OD solutions, showing a significant difference between these values [128]. Shi et al. (2013) synthetized hyaluronic acid-derivatized carbon nanotubes (HA-CNTs) and studied the cell viability of B16F10 cells with different concentrations of HA-CNTs, the PS hematoporphyrin monomethyl ether (HMME), and the composite HMME-HA-CNTs based on PDT and PTT effect. They used light irradiation of 532 nm (100 mW/cm^2^) with HMME and near-infrared 808 nm irradiation (1.4 W/cm^2^) with HA-CNTs. It was shown that HMME exhibited relatively small cytotoxicity to B16F10 cells, while the HMME with 532 nm laser group greatly enhanced cytotoxicity. In contrast, HA-CNTs exhibited a stronger inhibition than HMME. HA-CNTs are superior to HMME in terms of both their PDT and PTT efficiency [129].

The efficiency of CBM gave way to the design of nanocomposites, leading to the theranostic effect, which allows for the observation of a higher concentration of CBM loading PS in the tumor in relation to PS alone (metal-organic or organic). PS is often limited because of reduced water solubility, photostability, prolonged cutaneous photosensitivity, and low selectivity [130].

The advantages of using nanoplatforms in theranostic CBM designs have been widely studied [59,131]. Xie L. et al. (2016) designed a long circulation multifunctional albumin/Ce6 loaded evans blue/carbon nanotube-based delivery system (ACEC) that showed synergistic PDT and PTT along with efficient tumor ablation effects using red and IR light of 630 nm and 800 nm, respectively. In vivo fluorescence imaging and photoacoustic imaging of SCC7 tumor were developed and evidenced in the Ce6-only group that the signal was mainly distributed in the liver 8 h after treatment and disappeared 24 h after injection. The fluorescence images of the tumor regions gradually strengthened over time and reached their maximum point (T/M ratio = 7.83 ± 0.31) at 24 h post-treatment in the ACEC group and were gradually eliminated, indicating that ACEC circulated for a greater duration and accumulated more than the free Ce6 and albumin/Ce6. Compared to PDT or PTT alone, the combined phototherapy managed to damage the tumor and diminish the tumor without return [132]. Romero et al. (2021) designed theranostic nanocomposites based on GO with a surface modification of PEG-folic acid, rhodamine B, and indocyanine green (GO+RhodB, GO+PEGFA+RhodB, and GO+PEGFA+ICG). In addition to displaying red fluorescence spectra rhodamine B (as the fluorescent label), in vivo experiments were performed using GO to apply PDT and PTT in the treatment of Ehrlich tumors in mice using NIR light. The study was performed to obtain the highest concentration of GO in the tumor as a function of time (time after intraperitoneal injection). The obtained time was used to treat the tumor by PDT/PTT. Tumor volume control shows that the GO+PEGFA+ICG group has the lower “R factor” ((V-V_o_/V_o_) normalized by day’s number of follow-up) related to only the indocyanine group. Compared to PDT (only indocyanine PS) and PTT alone (only GO), the theranostic therapy diminished the tumor without recurrence (see Figure 16) [72].

Hua et al. (2018) synthetized C-DOTs via hydrothermal reaction of m-phenylenediamine and L-cysteine. The cytotoxicity of C-DOTs and the PS PpIX in HeLa cells were evaluated. The dark toxicity and the PDT efficacy of C-DOTs-PpIX and PpIX on the mentioned cells were confirmed. Free PpIX and C-DOTs-PpIX elicited no obvious cytotoxicity toward the cells after incubation for 24 h in the dark, indicating their excellent cytocompatibility. The PDT effect based on photoexcitation was studied and C-DOTs-PpIX showed more efficient cell-killing ability than free PpIX as the cell viability of C-DOTs-PpIX group was much lower than that of the free PpIX group [133].

Thus, nanoplatforms in theranostic CBM show advantages over organic and metal-organic PS due to their greater effectiveness and greater possibility of altering the molecule, thus increasing stability and biological availability.

## 4. Toxicity and Immune Response of CBM

The interaction of any nanoparticle with biological molecules depends on its physicochemical characteristics. Given the many advantages of CBM, it is also necessary to highlight the possible effects of toxicity. Composition, size, shape, charge, functionalization, and aggregation are some examples of these properties that must be studied and understood to avoid toxic effects. Nanomaterials, especially carbon-based, appear in different shapes and structures, such as particles, tubes, fibers, and films, and this variation directly affects their kinetics and delivery to the target cell (see Figure 17). Consequently, the route of transport in the environment is also affected [134].

ROS production is one of the essential mechanisms of toxicity of these nanomaterials and can lead to a chain of reactions ranging from inflammation and oxidative stress to protein denaturation and cell death [136]. This generation of ROS can reduce the membrane potential of mitochondria, causing damage to the cell and can react with fatty acids, causing lipid peroxidation. In the nucleus, CBM can also cause genotoxicity by interacting with DNA. These processes are the main factors related to toxicity and cell death caused by carbon-based nanomaterials [137].

The small size of nanomaterials enables them to cross biological barriers, including membranes, which can lead to cell damage (see Figure 18) [134]. Smaller CBM have shown greater oxidative capacity and have caused more oxidative damage to alveolar epithelial cells than the bigger ones [138]. Regarding graphene-based nanomaterials, GO, for example, due to their size, can cross cell membranes, which can damage them and promote cytoplasmic leakage with the generation of ROS [139]. Furthermore, when the surface area of nanoparticles is large, there is an increase in their binding capacity with other compounds on their surface, increasing their toxic effect in the biological environment. The bigger the surface area of CBM, the greater their oxidative potential, including the ability to damage DNA [140].

Understanding the characteristics of CBM is essential to improving their biocompatibility. For example, carbon nanotubes are hydrophobic and can aggregate in the blood, and several strategies have been adopted to improve their delivery in the biological environment via the intravenous route [142]. Typically, in experiments for PDT and PTT, a concentration without dark toxicity is used. Thus, cell death must be due to the interaction of light with the compound (be it photosensitive or photothermal).

Different ligands should reduce cytotoxicity. In PEG-functionalized GO, no significant changes were observed in the Zebrafish model [143] compared to several already shown effects of GO without PEG. This functionalization improves its solubility and compatibility and has also reduced toxic effects in mice [144]. Intravenous administration of amine-functionalized GO or only GO in mice proved the thrombogenic capacity of graphene, where functionalized graphene showed low toxicity [145]. Functionalization also changed the toxicity of single-wall carbon nanotubes, showing more biocompatibility for the presence of PEG since PEGylation alters its cytotoxic potential and has improved its excretion in animal tests [142].

For example, nano-GO modified with PEG ranging in concentration between 0 and 50 µg/mL maintain viability at more than 90% in the dark, which indicates low toxicity. However, when illuminated, cell viability drops to less than 60% for a light dose of 19.2 J/cm^2^ [61]. Even when r-GO is linked with other ligands besides PEG (such as molybdenum sulfide and manganese dioxide), varying the concentration from 25 to 200 µg/mL, there is excellent viability in the dark, showing low toxicity [77]. When illuminating, however, viability reduces by about 70%, as desired for PDT and PTT treatments.

Functionalization can also help increase specificity since only diseased cells will receive the nanoparticles, reducing toxicity in healthy tissues. Another possibility to improve CBM is their immobilization in a polymer matrix, preventing penetration into the cell and sealing its edges. However, these matrices must be very well studied concerning their biocompatibility for human application [139].

With the combined use of CBM with other drugs and specific ligands, it is possible to make complex molecules such as oxidized single-walled carbon nanotubes alloys with chemotherapy (docetaxel) and folic acid. Even when there is the presence of the chemotherapeutic agent in low concentrations, it can show low toxicity without illumination. At a concentration of 0.01 µM, this compound already shows a reduction in tumor cell viability of more than 20% in the dark. When illuminated, this reduction reaches more than 60% [146]. Therefore, understanding the toxicity and parameters involved in this process can generate more effective treatments, especially under lighting. When we analyzed the in vivo results of this study, there was no toxicity, showing that the inhibitory effect is only for the tumor and increases under irradiation.

It is necessary to understand the process of elimination and metabolism of CBM in the human body to be used in medicine. If they are not eliminated or accumulate in some vital organ, they can pose a health risk and, therefore, studies in this area are critical [142].

The cytotoxicity results of CBM are often contradictory. Not only because the physical-chemical characteristics can influence these results, but the synthesis process itself and the presence of metallic impurities can also influence them. Thus, it is still unclear what plays a central role in the immune response and the toxicity of these nanomaterials [147].

These results are found especially for graphene, whose structural properties directly influence its interaction with cells. Chemical oxidants and reducing agents used in graphene synthesis can result in organic contamination and metallic impurities, resulting in cell damage and increased toxicity. Carbon nanotubes are considered genotoxic, and this characteristic can be explained both by the fibrous property of these materials and the presence of impurities present in their composition [148]. The use of green synthesis can reduce their cytotoxicity and be a possibility in biomedical applications [139].

It should be noted that the prediction of in vivo toxicity from in vitro data does not necessarily reflect reality due to the different conditions of the cellular environment and the complex organism. However, understanding these processes can go a long way in controlling these toxic effects. In vivo studies can provide information about complex parameters such as metabolism and evaluation of the chronic effect, which can contrast with punctual in vitro results. The contradiction in the toxicity of the same compound may also result from these differences [134]. The poor dispersion of CBM in water also plays an essential role in the toxicity and reproducibility of both in vitro and in vivo assays [149].

CBM absorb over a broad region of the visible spectrum and can directly interfere in assays performed by fluorescence or absorption. Thus, the inconsistency concerning toxicity can still result from a technical inconsistency in the reading of toxicity tests with the use of CBM. Therefore, it is necessary to characterize these materials, especially regarding the reproducibility of toxicological tests. Ensuring the reproducibility of the technique itself ensures that better characterized and less toxic effects are found, which is significant as they are easily modified. Removing metallic impurities, adding binders on its surface, or increasing its dispersion are examples of processes with apparent effectiveness in producing less toxic materials [147].

Graphene and carbon nanotubes have different geometries. While the last one is shaped like a tube, as the name implies, the first one is formed by flat sheets. Thus, the interaction of these materials with the biological environment must happen through different mechanisms and, therefore, result in different responses, including specific toxicity and immunological effects [142]. Dose, interaction time, cell type, and animal lineage can influence the toxicity result [139].

The route of administration, such as oral, intraperitoneal, intravenous, or ocular, also interferes with the toxicity of CBM [150]. For example, studies have shown few toxic effects of graphene-based compounds on intraocular administration, with minimal effects on morphology and cell viability [151]. In comparison, studies have shown that graphene can induce chronic toxicity in mice from oral or systemic administration [139]. Pulmonary toxicity with multiple effects has also been reported, depending on the dose of graphene administered [144]. This also affects the toxicity of carbon nanotubes. Lung exposure leads to different effects, including genotoxic and intravenous injection that has resulted in the accumulation of carbon nanotubes in vital organs. However, these responses are mainly dependent on dose, functionalization, and physicochemical properties [152].

Graphene toxicity was shown to be dose-dependent, as it had significant effects at high doses in mice (such as inflammation and pulmonary edema), whereas it exhibited little or no effect at low and medium doses. Therefore, it is necessary to avoid general conclusions regarding these CBM. Understanding the biological interaction of the compounds (functionalized or not) can guarantee better results and may be the key to them being considered non-toxic [144].

Different compounds can interact with immune cells and trigger a response. As for toxicity, carbon-based compounds can act differently depending on their properties, such as size, shape, dispersion, and different functionalization [147].

Macrophage tests are the most used to study the immune response of these materials. Studies of cell uptake and viability and induction of an inflammatory response are some examples of these tests. Especially in macrophages, CBM have been exhibiting detrimental effects. However, the nanoparticle size may be definitive for this issue since, for example, short and long carbon nanotubes can show different effects. Another group of essential cells for the immune response is lymphocytes. CBM have been studied and both a positive and negative effect on modulating the cellular response of lymphocytes were determined, also showing the toxicity of these materials [147].

There are several in vitro toxicity studies with carbon nanotubes, especially in lung and skin cells and cells related to the immune system. Most have been showing decreases in cell proliferation, which can reduce the adhesion of carbon nanotubes, generating a series of reactions that will damage the membranes and cause cell death. Other studies have shown the potential for DNA damage and genetic alterations with carbon nanotubes [152].

In addition to the individual in vitro responses, in vivo tests allow studying the global immune response of compounds in the living organism from a systemic response. Studies with alveolar macrophages, inflammatory induction in the lungs, airway epithelial cells, and mast cells are some examples of what has been studied with CBM. Systemic activation of the immune response and the complement system are also possible with in vivo studies. Carbon nanotubes, for example, can activate this systemic response [147].

The use of high doses, the lack of standardization, exposure to professionals, and the effect on the environment are some examples of the origin of these problems. The development of regulations, whether in the production or disposal of these materials, is an essential step in reducing the adverse effects of nanomaterials and has been a concern of the global scientific community [134]. The influence of graphene on plantations has shown that it can induce adverse effects, including on plantation roots, being dose-dependent. The structure can also be a determining factor, as few layers did not significantly affect plant growth rates. Furthermore, high concentrations of graphene can affect microorganisms in the environment, including water, affecting plant growth and the general biome. Therefore, understanding toxicity must be a constant concern for those working with these compounds [144].

Although CBM have wide applications, before applying these materials for any research it is necessary to study and understand their intrinsic toxic effects, mainly because they often allow for structural modifications and subsequent reductions in toxicity, making them even more attractive for use in nanomedicine.

## 5. Conclusions

Based on the compilation presented in this review, it is evident that a wide range of carbon-based materials is being used within phototherapy and synergetic therapies to battle against cancer disease. Graphene oxide, reduced GO, graphene quantum dots, and carbon dots have been studied by several researchers showing excellent results for tumor destruction. Nevertheless, it is necessary to define standard guidelines for preparing and applying this type of material for therapeutic purposes. Researchers must employ in vivo and in vitro applications to humans. It is worth noting that the phototherapy field using carbon-based material is still in an emerging phase. Therefore, further exploration remains to be done until full high-quality implementation and commercialization can happen. A grand challenge is to develop a treatment with reduced dose-dependent toxicity to improve the care of cancer patients.

## Figures and Tables

**Figure 1 ijms-23-00022-f001:**
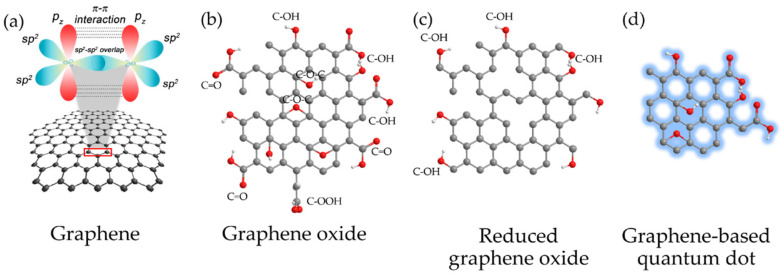
CBM: (**a**) Graphene with sp^2^-hybridized carbon atoms; (**b**) GO; (**c**) r-GO and (**d**) GQD. Reproduced from Ref. [50]. Copyright 2017 MDPI.

**Figure 3 ijms-23-00022-f003:**
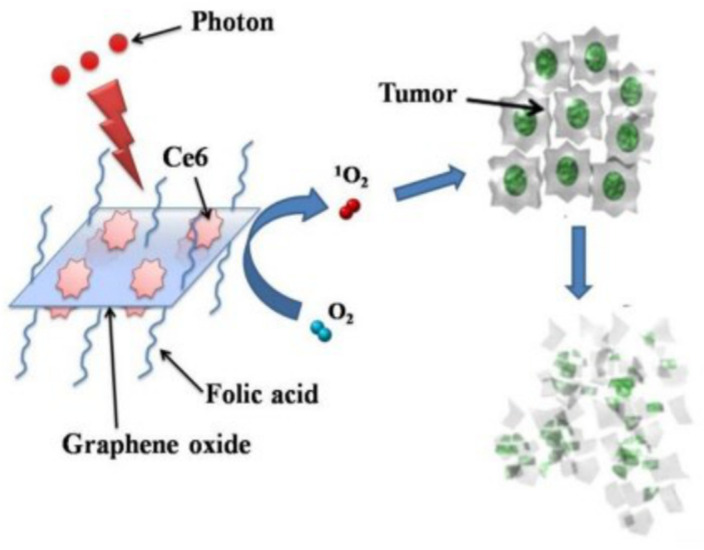
PS molecules of Chlorin e6 (Ce6) loaded by folic acid-conjugated GO for PDT applications in cells. Reproduced from Ref. [63]. Copyright 2012 PubMed Central.

**Figure 4 ijms-23-00022-f004:**
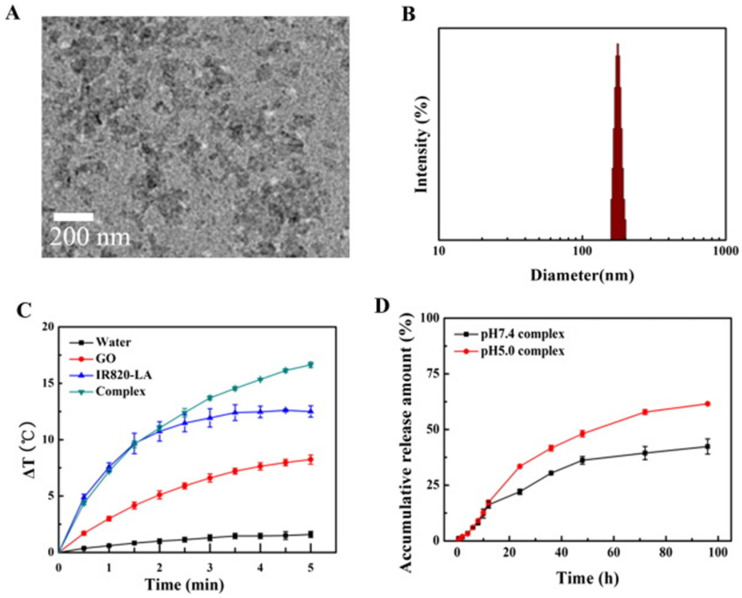
(**A**) TEM image of the composite; (**B**) Composite particle size distribution (DLS); (**C**) Photothermal effect curves of the composite, IR820-LA, GO, and water under a 660 nm laser (n = 3); and (**D**) In vitro DOX drug release of the composite. Complex* correspond to GO/DOX/IR820-LA composite. Reproduced from Ref. [67]. Copyright 2019 ELSEVIER.

**Figure 5 ijms-23-00022-f005:**
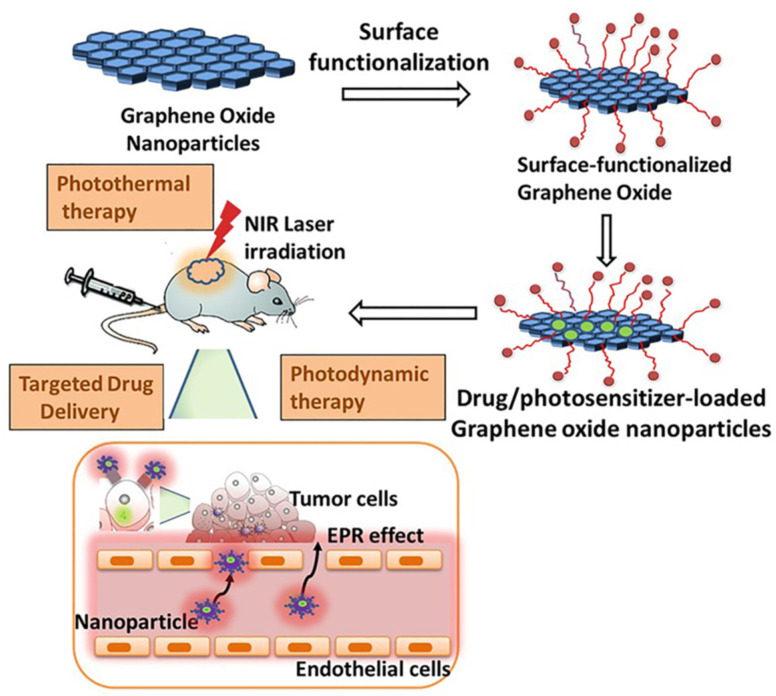
Procedure to apply GO-based hybrids/composites in PDT, PTT, and targeted drug delivery. EPR: enhanced permeability and retention. Reproduced from Ref. [58]. Copyright 2020 MDPI.

**Figure 6 ijms-23-00022-f006:**
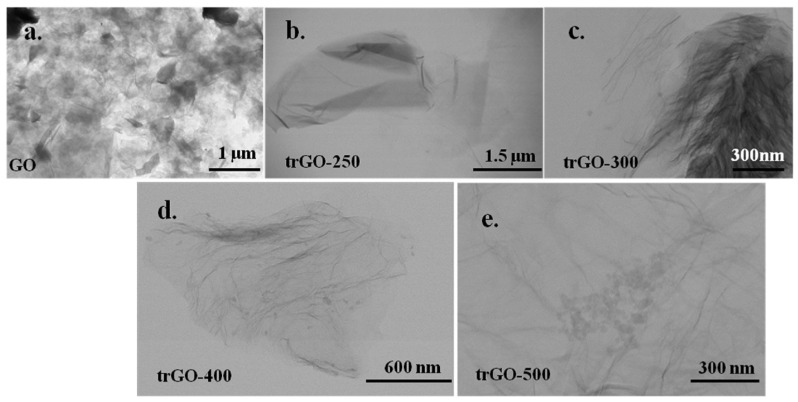
TEM images of (**a**) GO; and its reduction to r-GO at different temperatures: (**b**) 250 °C, (**c**) 300 °C, (**d**) 400 °C, and (**e**) 500 °C. Reproduced from Ref. [73]. Copyright 2020 ELSEVIER.

**Figure 7 ijms-23-00022-f007:**
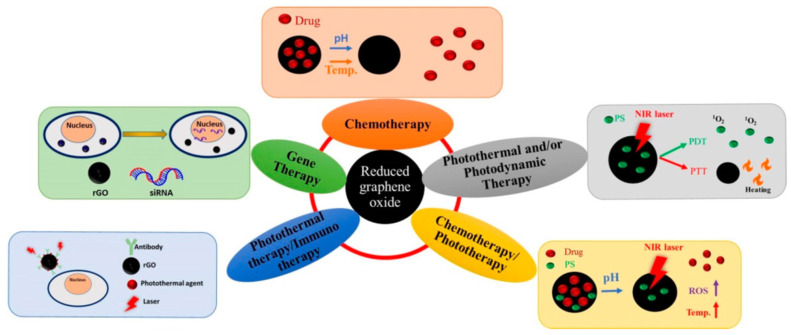
Applications of r-GO in PDT and PTT, chemotherapy/phototherapy, photothermal/immune therapy, gene therapy, and chemotherapy. Reproduced from Ref. [76]. Copyright 2021 MDPI.

**Figure 8 ijms-23-00022-f008:**
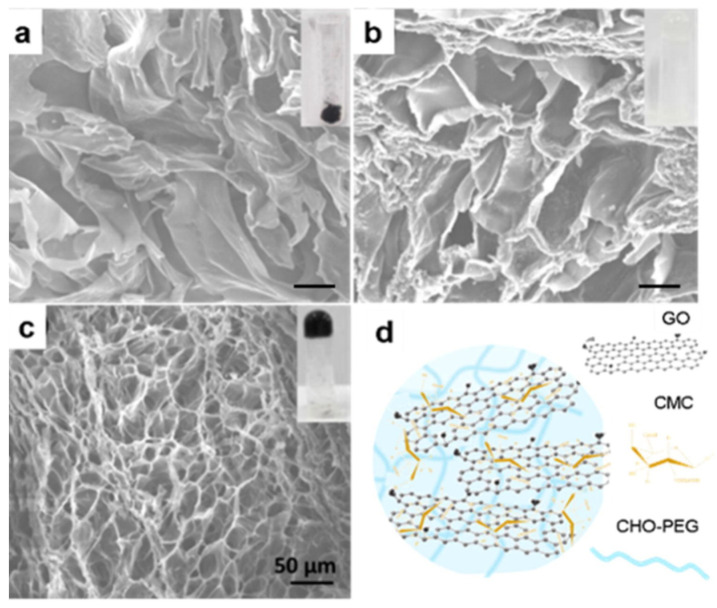
SEM images of: (**a**) CMC/r-GO powder; (**b**) CMC/CHO/PEG hydrogel; (**c**) CMC/r-GO/CHO/PEG hydrogel; and (**d**) illustration of CMC/r-GO/CHO/PEG hydrogel. The insets show photographs of the corresponding samples. Reproduced from Ref. [82]. Copyright 2019 ELSEVIER.

**Figure 9 ijms-23-00022-f009:**
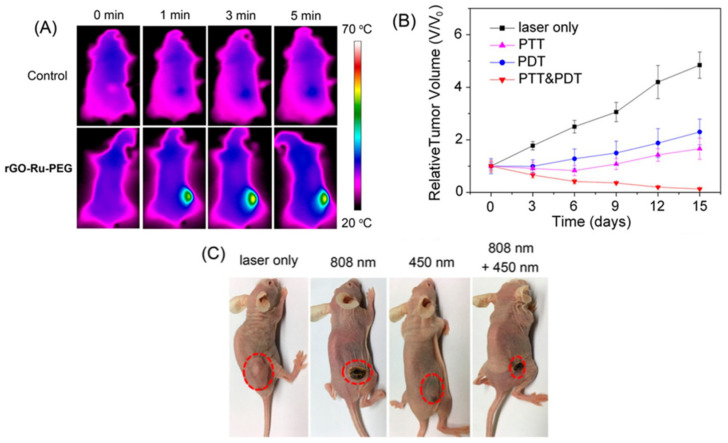
(**A**) IR thermal images of A549 tumor-bearing mice exposed to 808 nm laser for 5 min. (**B**) Tumor growth curves of different groups of A549 tumor-bearing mice (n = 5). (**C**) Photos of mice after various treatments taken on day 15. Reproduced from Ref. [86]. Copyright 2017 ACS.

**Figure 10 ijms-23-00022-f010:**
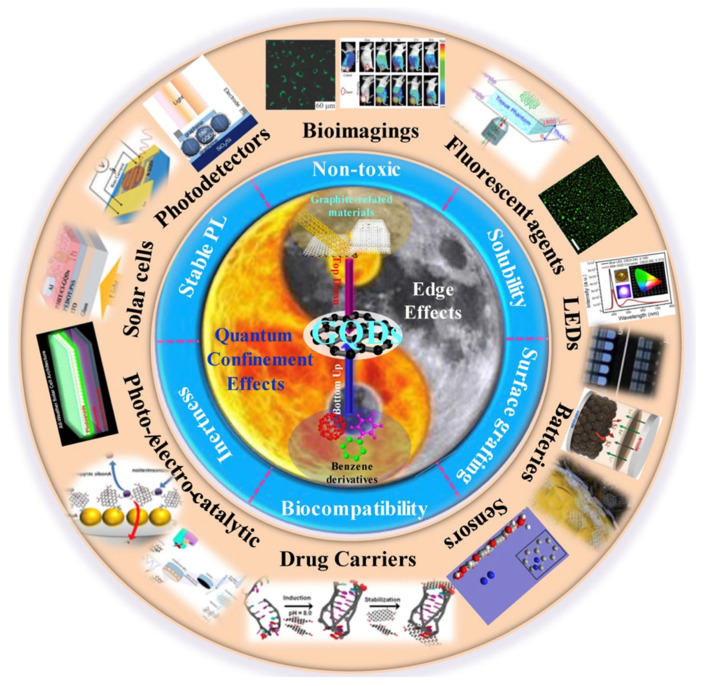
GQDs related inherent effects, preparation methods, properties, and applications. Reproduced from Ref. [53]. Copyright 2018 ELSEVIER.

**Figure 11 ijms-23-00022-f011:**
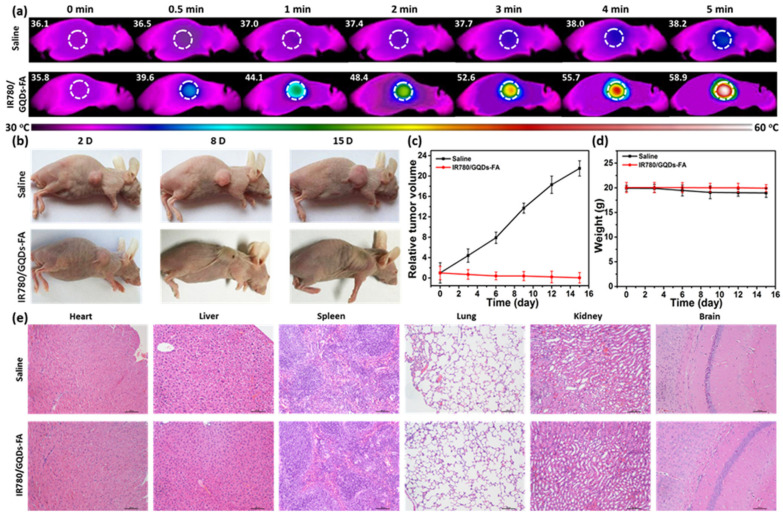
(**a**) Tumor thermal images of temperature variations after intravenous injection of saline and IR780/GQDs-FA with laser irradiation; (**b**) Photos of the tumor-bearing mice after treatments. Tumor volume (**c**) and body weight (**d**) curves of the tumor-bearing mice. (**e**) Histological evaluation of tissues from the mice treated with saline and IR780/GQDs-FA. Each organ was sliced for hematoxylin and eosin (H&E) staining. Reproduced from Ref. [102]. Copyright 2017 ACS.

**Figure 12 ijms-23-00022-f012:**
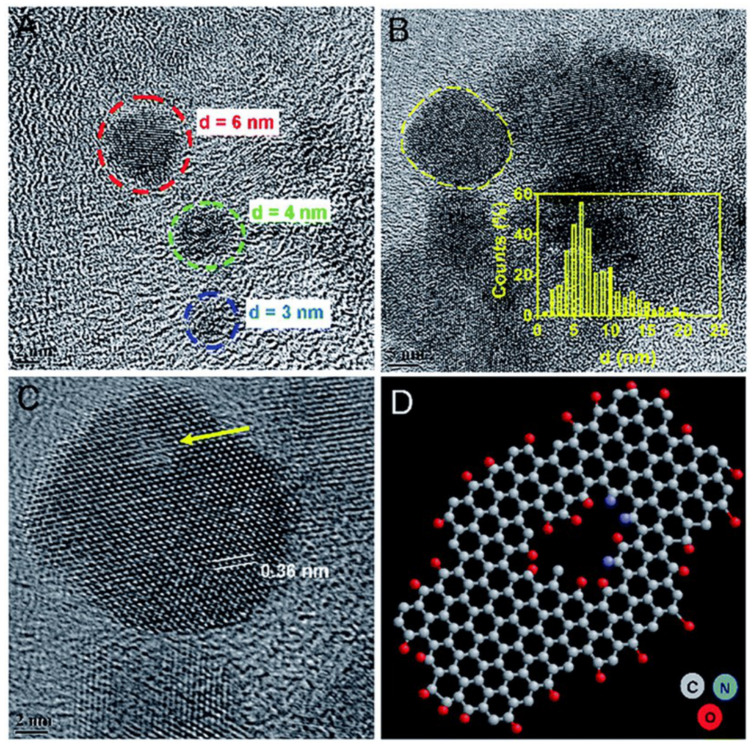
Morphological analysis of GQDs from withered leaves. TEM image of (**A**) ultra-small GQDs and (**B**) existence of GQDs in cluster form with inset showing size distribution. (**C**) HRTEM of a single crystalline GQD (marked in (**B**)) with a honey-comb-like structure of graphene with few basal/edge state-defects shown with an arrow. Inset here shows lattice spacing distance. (**D**) Ball and stick model indicating the defect in a typical structure of GQD with self-passivated functional groups. Reproduced from Ref. [105]. Copyright 2017 Royal Society of Chemistry.

**Figure 13 ijms-23-00022-f013:**
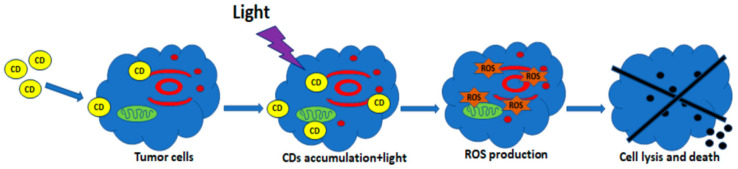
Schematic diagram of PDT. C-DOTs (CDs) penetrate the cell membrane and accumulate in the cytosol. Light irradiation actives and induces the production of ROS. Reproduced from Ref. [116]. Copyright 2021 MDPI.

**Figure 14 ijms-23-00022-f014:**
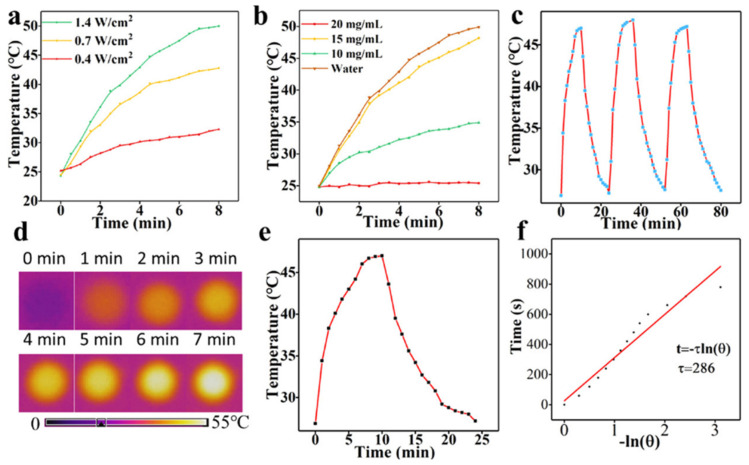
(**a**) Temperature curves of C-DOTs solution (20 mg/mL); (**b**) Temperature curves of different concentrations of C-DOTs solutions; (**c**) Photothermal profiles of C-DOTs solution with 3 irradiation cycles; (**d**) Thermal infrared images of C-DOTs solutions recorded after 7 min of irradiation; (**e**) Heating and cooling curves of C-DOTs solution; (**f**) Linear time data and −ln θ acquired from a cooling period of (**e**). 1.4 W cm^−2^; 15 mg/mL. Reproduced from Ref. [121]. Copyright 2019 ACS.

**Figure 15 ijms-23-00022-f015:**
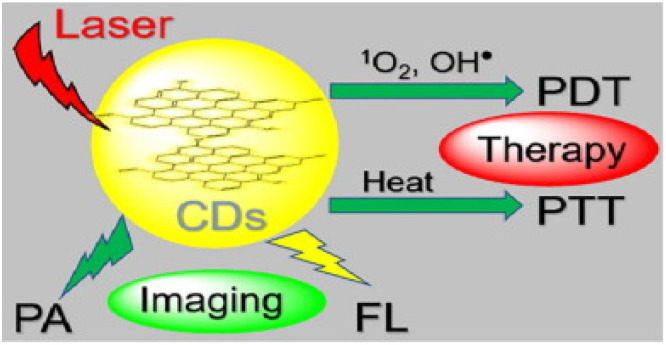
Lysosome targetable C-DOTS which can simultaneously generate ^1^O_2_, OH^●^, and heat under 635 nm laser irradiation. Reproduced from Ref. [123]. Copyright 2020 ELSEVIER.

**Figure 16 ijms-23-00022-f016:**
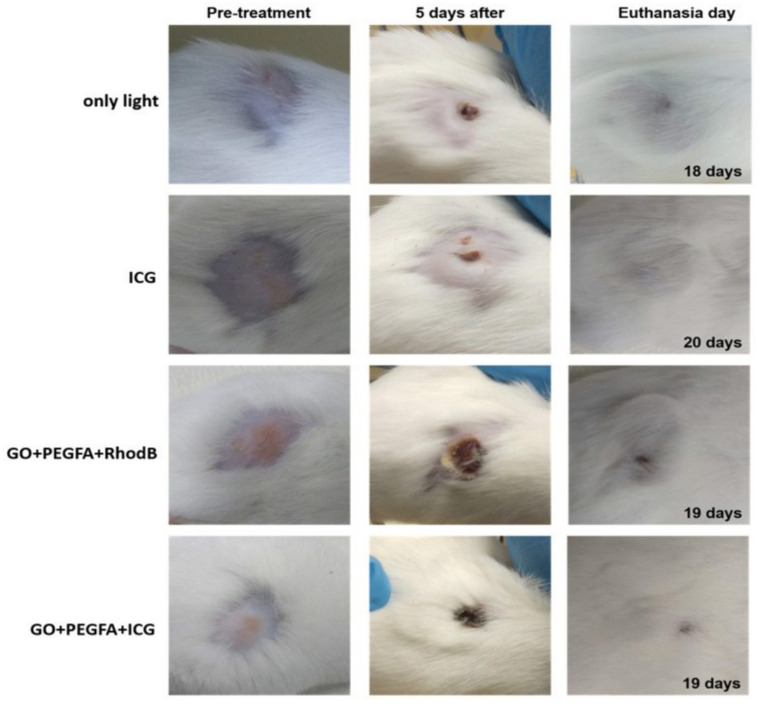
Representative mice images showing the sizes of tumors at pre-treatment day, 5th day, and euthanasia day under 808 nm, 1.8 W/cm^2^ for light, ICG, GO+PEGFA+RhodB, and GO+PEGFA+ICG mice groups. Reproduced from Ref. [72]. Copyright 2021 PMC.

**Figure 17 ijms-23-00022-f017:**
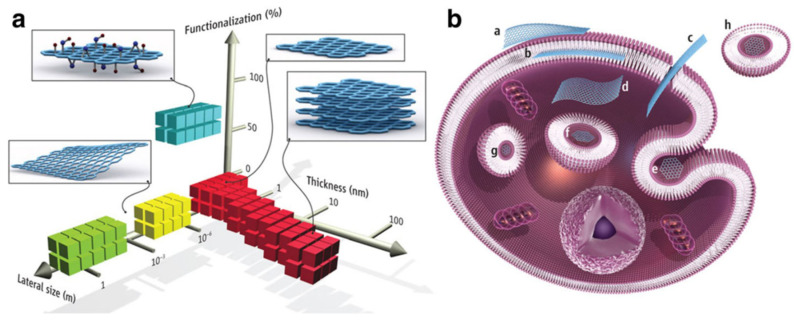
(**a**) CBM can be described by the dimensions and surface functionalization of the material (percentage of the carbon atoms in sp^3^ hybridization). Green squares represent epitaxially grown graphene; yellow, mechanically exfoliated graphene; red, chemically exfoliated graphene; blue, GO. (**b**) Possible interactions between CBM with cells (a) Adhesion onto the outer surface of the cell membrane. (b) Incorporation in between the monolayers of the plasma membrane lipid bilayer. (c) Translocation of the membrane. (d) Cytoplasmic internalization. (e) Clathrin-mediated endocytosis. (f) Endosomal or phagosomal internalization. (g) Lysosomal or other perinuclear compartment localization. (h) Exosomal localization. The biological outcomes from such interactions can be considered to be either adverse or beneficial, depending on the context of the particular biomedical application. Different CBM will have different preferential mechanisms of interaction with cells and tissues that largely await discovery. Reproduced from Ref. [135]. Copyright 2014 the American Association for the Advancement of Science.

**Figure 18 ijms-23-00022-f018:**
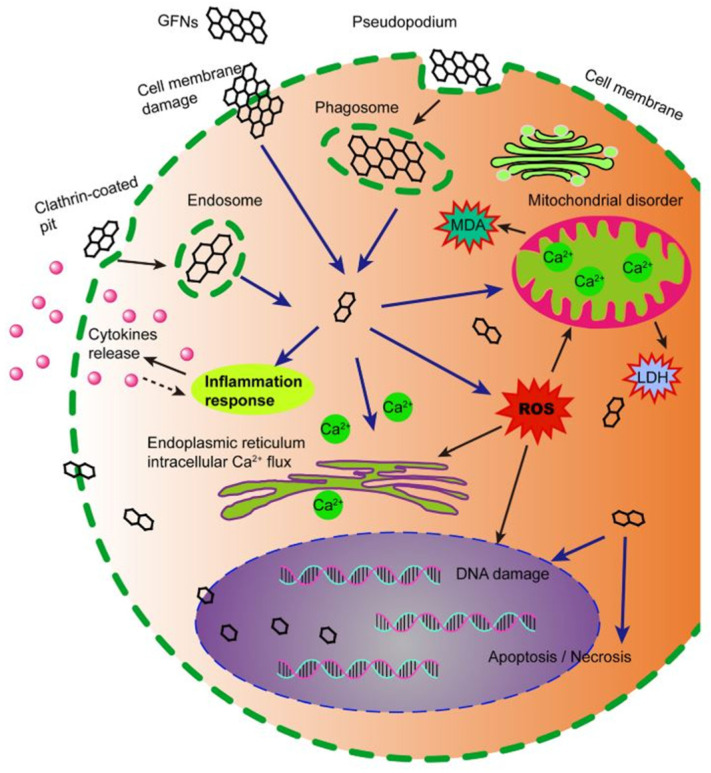
Schematic diagram showing the possible mechanisms of CBM cytotoxicity. CBM get into cells, which induces ROS generation, lactate dehydrogenase and malondialdehyde increase, and Ca^2+^ release. Subsequently, CBM cause kinds of cell injury, for instance, cell membrane damage, inflammation, DNA damage, mitochondrial disorders, apoptosis, or necrosis. Reproduced from Ref. [141]. Copyright 2016 Springer Nature.

## Data Availability

Not applicable.

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
