# Peer review of "Carbon-Based Materials in Photodynamic and Photothermal Therapies Applied to Tumor Destruction"

_ijms, 2021, doi:10.3390/ijms23010022_

Round 1

Reviewer 1 Report

In general, the manuscript is well written and comprehensible, with a few gramatical errors and common typos. Nevertheless, some sentences and paragraphs must be rephrased for clarification (see attachment). 

Only 9 figures, most of them somewhat generic (and some of them - I counted at least 4 - simply the graphical abstacts of the cited articles!!), for a 139-reference review paper (I don't know why you refer to it as a mini-review in the text...) is far too short in my opinion. 

Hence, more illustrative figures or schemes are needed in section 2, and some of these have to be more specific of the papers that are being referenced, including, for instance: some selected details on the preparation procedures and chemical structures of the PSs used in the modified CBM; a few more elaborate in vitro or in vivo studies carried-out with them; or some interesting SEM or TEM images of novel functionalized nanomaterials. I think this would make the manuscript more complete and far more eye-catching for the readers. 

Section 3, regarding the toxicity issues, is again quite generic, and could be improved extracting more specific aspects of some of the papers that are cited throughout section 2, at least in the cases where those issues were assessed, as well as comparing to the toxicity effects of 'normal' PDT and PTT, without the use of composite / hybrid agents based on CBM.

All corrections, suggestions and queries, either quite simple or more relevant, were added using the adobe acrobat reader editing tools and can be found in the revised manuscript file itself (see attachment).

Author Response

Dear Reviewer 1:

Reviewer 2 Report

Author wrote the review paper titled "Carbon-Based Materials in Photodynamic and Photothermal Therapies Applied to Tumor Destruction" covered some preparation procedure, applications in PDT and PTT. But, this review paper didn't include enough information. Author should give broader information about carbon based materials in PDT and PTT. The comments are below.

  1. Author should include general synthesis scheme for all graphene based materials mentioned in this review.
  2. Author should explain the advantages of carbon-based materials in PDT and PTT over  metal-organic or organic Photosensitizers.
  3. Author missed to cite several papers.
  4. ACS Nano 2011, 5, 9, 7000-7009 in applications in GO in PDT.
  5. Author should include TEM images of rGO
  6. Author should cite the paper "Photothermal and Photodynamic therapy reagents based on rGO-C6H4-COOH" RSC Adv., 2016, 6, 3748.
  7. Author should cite the paper " Graphene Quantum Dots in Photodynamic Therapy" Nanoscale Adv., 2020, 2, 4961-4967.
  8. There is a typo in citation 103. (missing 12).
  9. Author should cite the paper "Red-emission hydrophobic porphyrin structure carbon dots linked with transferrin for cell imaging" Talanta 217(2020) 121014.
  10. Authoer should cite the paper "Theranostic carbon dots with innovative NIR-II emission for in vivo renal-excreted optical imaging and photothermal therapy". ACS Appl.Mater. Interfaces 2019,11,5, 4737-4744.
  11. Please check the title of the review. The letter "a" in applied should be caps "A".

Author Response

Dear reviewer 2:

Round 2

Reviewer 2 Report

This review is much more polished than before. Please accept it as in  present form.